# Improvement of the Photoelectrochemical Performance of TiO₂ Nanorod Array by PEDOT and Oxygen Vacancy Co-Modification

**Bin Yang [1], Guoqiang Chen [2], Huiwen Tian [3,\*] and Lei Wen [1,\*]**

[1] National Center for Materials Service Safety, University of Science and Technology Beijing, Beijing 100083, China; binyang@ustb.edu.cn

[2] Beijing General Research Institute of Mining & Metallurgy Group, Beijing 100160, China; ccggqq871020@126.com

[3] Key Laboratory of Marine Environmental Corrosion and Bio-fouling, Institute of Oceanology, Chinese Academy of Sciences, Qingdao 266071, China

\* Correspondence: tianhuiwen1983@foxmail.com (H.T.); wenlei@ustb.edu.cn (L.W.); Tel.: +86-010-62333132 (H.T.); +86-0532-82898832 (L.W.)

**Abstract:** In this study, oxygen vacancy modified TiO₂ nanorod array photoelectrode was prepared by reducing hydrogen atmosphere to increase its free charge carrier density. Subsequently, a p-type conductive poly 3,4-ethylenedioxythiophene (PEDOT) layer was deposited on the surface of oxygen vacancy modified TiO₂, to inhibit the surface states. Meanwhile, a p-n heterojunction formed between PEDOT and TiO₂ to improve the separation of photo-induced carriers further. The photocurrent of TiO₂ nanorod array increased to nearly 0.9 mA/cm$^2$ after the co-modification under standard sunlight illumination, whose value is nearly nine times higher than that of pure TiO₂ nanorod array. Thus, this is a promising modification method for TiO₂ photoanode photoelectrochemical (PEC) performance improving.

**Keywords:** oxygen vacancy; polymeric composites; photoelectrochemistry; co-modification; solar energy conversion

---

## 1. Introduction

TiO₂ has been widely investigated in the past few decades since Fujishima and Honda first reported its potential in the fields of photocatalysis and photoelectrochemistry in 1972 [1]. The theoretical limited photocurrent densities of anatase and rutile TiO₂ are 1.1 mA/cm$^2$ and 1.8 mA/cm$^2$ under solar light illumination, respectively. [2] Limited by its low solar light utilization rate and high photo-generated carrier recombination rate, many modification methods have been researched, such as metal doped [3], non-metal doped [4], and construct heterojunction [5]. Several elements have been introduced into TiO₂, such as Fe [6], S [7], and N [8]. Metal and non-metal doping could narrow the bandgap, extend the light absorption range and increase the charge carrier density to improve its photocatalysis performance. However, the introduction of heterogeneous atoms is likely to cause asymmetric doping or impurities, which would serve as recombination centers for the photo-generated electrons and holes, therefore reducing the PEC performance. Many previous research works showed that the formation of surface oxygen vacancy [9–13] could increase the charge carrier density of the semiconductor to improve its PEC performance. Wang et al. [14] obtained a yellowish ZnO with a narrowing band gap by introducing the oxygen vacancies into ZnO crystal, which increased the free charge density of the ZnO, so that the transfer process of the photogenerated charges became feasible.

Polymer organic semiconductors with good film-forming properties, high conductivity, high visible light transmittance and excellent stability are widely used in the field of photoelectrode modification. Park et al. [15] used a blend of 100 nm $TiO_2$ scattering particles in PEDOT:PSS (poly 3,4-ethylenedioxythiophene:poly styrenesulfonate) solution to fabricate transparent electrode films. When utilized in an organic photovoltaic device, a power conversion efficiency of 7.92% was achieved. Sakai et. al. [16] assembled PEDOT and $TiO_2$ layer-by-layer to switch electric conductivity in response to ultraviolet and visible light. PEDOT is a promising material to modify the $TiO_2$ photoanode to improve its PEC performance [17–20].

Therefore, in this work, we prepared oxygen vacancy modified $TiO_2$ nanorod array photoanode with high charge mobility capacity. Then, a p-type PEDOT layer was covered on the surface of oxygen vacancy modified $TiO_2$ photoanode to inhibit the undesirable surface state and construct a p-n heterojunction to accelerate the separation capacity of photo-generated carriers [5].

## 2. Results and Discussion

The XRD (X-ray Diffraction) patterns of series samples were shown in Figure 1, all diffraction peaks of the prepared three photoelectrodes can be indexed as rutile-type and anatase-type $TiO_2$ (JCPDS No. 21-1276, JCPDS No. 21-1272) [21,22]. The characteristic diffraction peaks at $2\theta = 36.08°$, 54.32, 62.74, and 69.78° corresponded to the (101), (211), (002), and (112) crystal planes of rutile-type $TiO_2$, and the XRD peaks at $2\theta = 63.68°$ corresponded to the (204) crystal planes of anatase-type $TiO_2$. The other characteristic diffraction peaks at $2\theta = 26.57°$, 37.76°, 51.75°, and 65.74° corresponded to the (110), (200), (211), and (301) crystal planes of $SnO_2$ (JCPDS No. 46-1088), which caused by the fluorine doped tin oxide (FTO) conductive glass. So, the prepared $TiO_2$ nanorod array included rutile phases and little anatase phases. The $TiO_2$ nanorod array preparing method in this work was referred to in Liu's work [17]. The vanished peaks for anatase and rutile $TiO_2$ at 25.4° and 27.4° on the XRD curves maybe attributed to the crystal face inhibition effect of the oriented growth nanorod structure, whose results are similar to Liu's work [17].

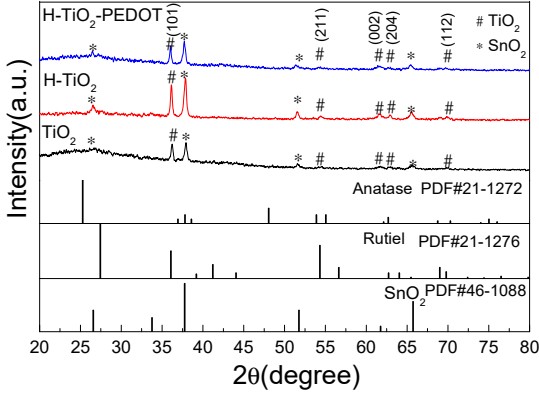

**Figure 1.** XRD patterns of $TiO_2$, H-$TiO_2$, and H-$TiO_2$-PEDOT.

The SEM technique was employed to observe the surface morphologies of the series samples, and the results are shown in Figure 2. As presented in Figure 2A–C, both $TiO_2$, H-$TiO_2$, and H-$TiO_2$-PEDOT appear to have a distinct nanorod structure. The cross-section image of H-$TiO_2$-PEDOT shown in Figure 2D reveals that the $TiO_2$ nanorod is growing vertically on the FTO substrate. The nanorods are tetragonal in shape with square top facets, the expected growth habit for the tetragonal crystal structure. The nanorods are nearly perpendicular to the FTO substrate. After 8 h of growth, the average diameter and length, as determined from SEM, were 90 ± 20 nm and 1 ± 0.2 μm, respectively. The peaks of (101) crystal planes for rutile and (204) for anatase $TiO_2$ can be clearly observed in the HRTEM image inset in Figure 2D, which is in agreement with the XRD results. Meanwhile, PEDOT layer can be observed at the edge area of $TiO_2$ nanorod. Elements distribution

of H-TiO$_2$-PEDOT were tested by STEM and STEM-EDS mapping. The STEM mapping shown in Figure 2E reveals the uniform distribution of Ti, O, and S element on the surface of the nanorod, where the S element corresponding to the PEDOT deposition layer. This result indicates that PEDOT layer was successfully deposited on the surface of H-TiO$_2$ photoelectrode.

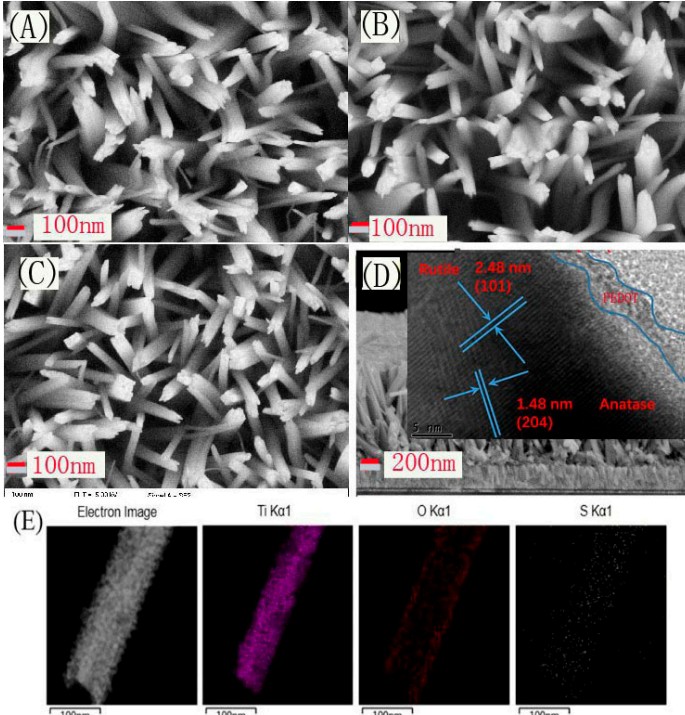

**Figure 2.** SEM images of (**A**) TiO$_2$, (**B**) hydrogen treated TiO$_2$ (H-TiO$_2$), (**C**) PEDOT modified hydrogen treated TiO$_2$ (H-TiO$_2$-PEDOT) and (**D**) cross-section image of H-TiO$_2$-PEDOT. Insert is the HRTEM image of H-TiO$_2$-PEDOT. (**E**) STEM mapping of H-TiO$_2$-PEDOT.

Similar results could be observed on EDS mapping (Figure 3), in which, the O, Ti, and Sn element corresponding to TiO$_2$ nanorod and FTO substrate were evenly distributed throughout all the H-TiO$_2$-PEDOT photoelectrode, besides, C and S elements could be observed simultaneously, which is corresponding to the STEM mapping showed in Figure 2E.

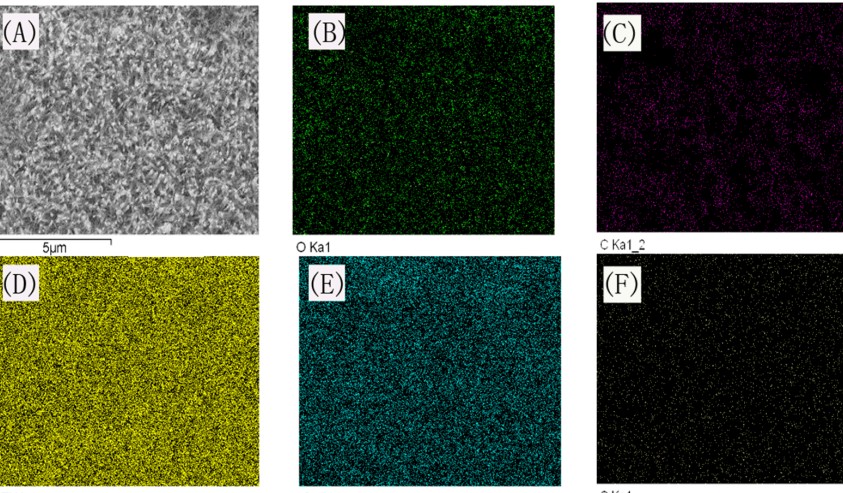

**Figure 3.** EDS mapping of H-TiO$_2$-PEDOT. (**A**) Scanning area, (**B**) O Element, (**C**) C Element, (**D**) Ti Element, (**E**) Sn Element, (**F**) S Element.

To determine the surface composition and chemical states of the series samples, high-resolution XPS spectra of O 1s and S 2p were used (see Figure 4). The characteristic peaks at 529.8 eV and 530.8 eV correspond to the lattice oxygen ($O_{lat}$) and the vacancies of O element ($O_{def}$). Compared to $TiO_2$, the peak area of $O_{def}$ in H-$TiO_2$ was enhanced after the hydrogen treatment, indicating the increase of oxygen vacancies from 34.2% to 43.77%, which might improve the PEC performance [16]. Oxygen vacancy concentration refers to the proportion of oxygen vacancy peak area to the total oxygen peak area. Because XPS can only read the distribution of surface elements, the peak area of oxygen (O 1s, $O_{lat}$, $O_{def}$) becomes smaller after PEDOT loading, but the relative content is credible. Next, the peak area of $O_{def}$ in H-$TiO_2$-PEDOT was reduced further after PEDOT deposition, which can be ascribed to the protection of PEDOT layer. The S was observed in the XPS spectra of H-$TiO_2$-PEDOT indicating PEDOT was introduced successfully, which corresponds to the result of the XPS survey spectra shown in Figure 4C. Because of the low loading amount of PEDOT, noises can be found on the XPS S2p curve. The characteristic peaks of Ti 2p did not shift after the hydrogen treatment, and the deposition of PEDOT (Figure 4D) indicated that the unique $TiO_2$ nanorod structure was preserved.

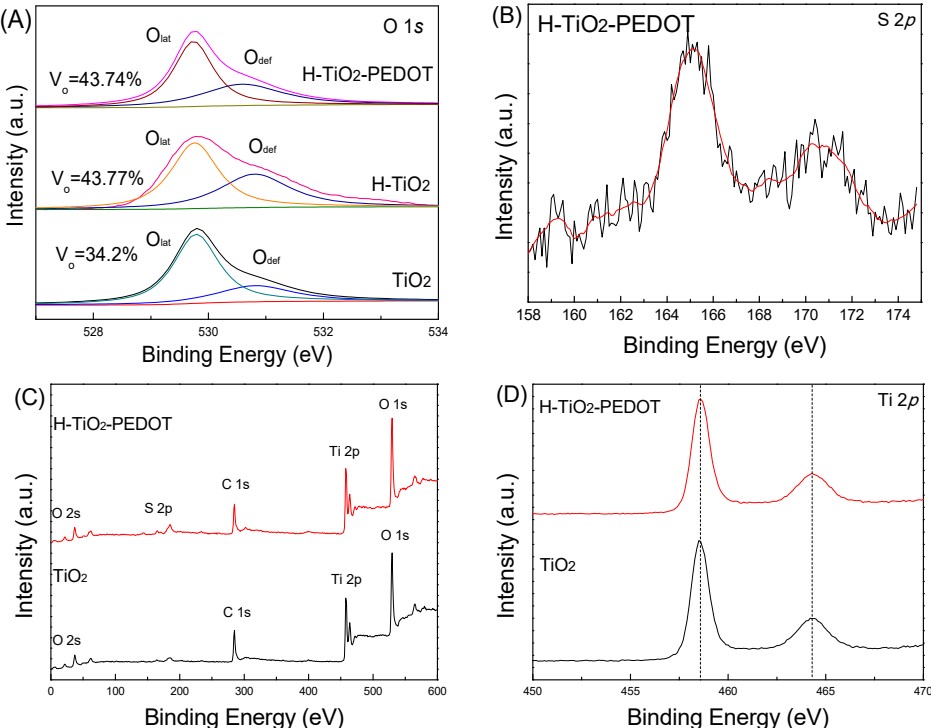

**Figure 4.** High-resolution XPS spectra of O 1s (**A**) and S 2p (**B**) of the $TiO_2$, H-$TiO_2$ and H-$TiO_2$-PEDOT. XPS survey spectra (**C**), high-resolution XPS spectra of Ti 2p (**D**) of the $TiO_2$ and H-$TiO_2$-PEDOT.

The PEC performance results of series samples are presented in Figure 5. Figure 5A is the current density-time curve of the series of electrodes, and Figure 5B is the current density-voltage curve of the series of electrodes. The current density-voltage curve shows that the current density of the $TiO_2$ sample at zero bias (vs. Ag/AgCl) is about 0.07 mA/cm$^2$. The current density of H-$TiO_2$ sample at zero bias is about 0.27 mA/cm$^2$. The current density of H-$TiO_2$-PEDOT sample at zero bias is about 0.33 mA/cm$^2$. In the voltage range from −0.5 to 0.5 V, the photocurrent density of sample H-$TiO_2$-PEDOT is higher than that of sample H-$TiO_2$, and the PEC performance of pure $TiO_2$ nanorod array is the worst. Figure 5C is the impedance data of each sample in the absence of light. The arc radius of pure $TiO_2$ is the largest, corresponding to the largest impedance. The arc radius of H-$TiO_2$ is the smallest, corresponding to the smallest impedance. After PEDOT deposition, the arc radius of H-$TiO_2$-PEDOT become larger because of the impedance of PEDOT. After oxygen vacancies modification, the arc radius and impedance of the obtained H-$TiO_2$ sample decreases. PEDOT conductive layer coated on

the hydrogen treated TiO$_2$ photoelectrode make the arc radius of the obtained H-TiO$_2$-PEDOT further smaller, indicating a smaller impedance of this sample.

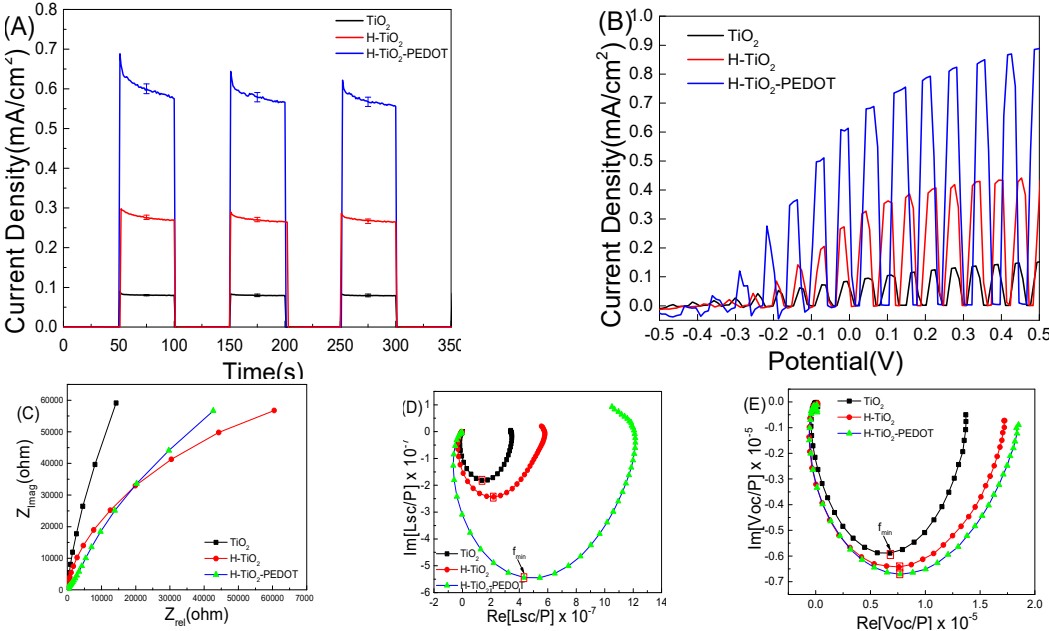

**Figure 5.** The current density-time curves (I-T) (**A**), the current density-potential (I-V) (**B**), AC impedance spectroscopy (EIS) (**C**), intensity modulated photocurrent spectroscopy (CIMPS) (**D**) and intensity modulated photovoltage spectroscopy (CIMVS) (**E**) of series photoelectrodes.

Figure 5D shows the CIMPS data of each sample under a monochrome light LED-365 nm with a 5% amplitude. The electron migration time of the sample can be obtained by converting the frequency of the minimum imaginary component into Equation (1), which is shown in the Experimental Section. Electron migration time represents the sum of the photogenerated electron time from excitation to the back electrode FTO and the time of photogenerated holes oxidation of the electrolytes in the electric double layer. Figure 5E is the CIMVS data in the same testing conditions. The electronic lifetime can be obtained by introducing the obtained frequency into Equation (2).

From the calculation results shown in Table 1, the electron migration time decreases obviously after hydrogen treatment. However, hydrogen treatment also introduces defects in the surface and bulk phase, which increases the probability of secondary recombination to reduce the lifetime of photogenerated electron holes. After the PEDOT conductive layer deposition, the surface state cannot be oxidized by air, meanwhile, p-n junction can be formed between TiO$_2$ and PEDOT thin film. The formation of p-n junction electric field accelerates the separation of photogenerated electron holes and reduces the electron migration time. The charge collection efficiency of these three samples was also calculated and the calculation process is shown in Equation (3). H-TiO$_2$-PEDOT photoanode shows a 37.71% charge collection efficiency whose value is higher than that of TiO$_2$ and H-TiO$_2$, indicating that more real hot carriers can be used in the PEC process.

**Table 1.** The calculated data through CIMPS and CIMVS results.

|  | $f_{min}$(CIMPS) | $t_r$ | $f_{min}$(CIMVS) | $t_{rec}$ | η (%) |
|---|---|---|---|---|---|
| TiO$_2$ | 172.24 | 0.924499 | 154.22 | 1.032523 | 10.46 |
| H-TiO$_2$ | 673.58 | 0.236402 | 536.63 | 0.296733 | 20.33 |
| H-TiO$_2$-PEDOT | 845.47 | 0.18834 | 526.63 | 0.302367 | 37.71 |

IPCE of the series of electrodes were tested and the results are shown in Figure 6A. It can be seen that the photoelectric conversion efficiency of hydrogen-reduced TiO$_2$ is significantly higher

than that of non-reduced TiO$_2$. After loading PEDOT on the photoelectrode, the H-TiO$_2$-PEDOT electrodes reducing surface state have more than 60% photoelectric conversion efficiency. Figure 6B presents the ultraviolet-visible diffuse reflectance result of the series photoanodes. It can be seen that the absorption band edge of pure TiO$_2$ is about 400 nm, because of anatase (band gap 3.2 eV) and rutile (band gap 3.0 eV) mix phase. After hydrogen treatment, an indicated absorption can be found from 400 nm to 600 nm, because of the oxygen vacancy energy level formed on the top of the TiO$_2$ valance band. The light absorption capacity of oxygen modified TiO$_2$ nanorod array did not change after the PEDOT outer layer loading. Comparing with Figure 6A, there is no photocurrent response of H-TiO$_2$-PEDOT photoanode in the wavelength area from 400 nm to 600 nm, indicating that there is no IPCE contribution from oxygen vacancy surface energy level. Figure 6C is a photocurrent-time curve measured continuously for 4 h under 0.5 V (vs. Ag/AgCl) external bias voltage. After 4 h continuous illumination, the photocurrent generating by H-TiO$_2$-PEDOT photoanode decays less than 10% of the initial value, showing acceptable stability. Meanwhile, the oxygen and hydrogen evolution performance were tested during the PEC stability testing for 4 h, and the result shown in Figure 6C indicate that the H-TiO$_2$-PEDOT photoanode can completely split pure water into hydrogen and oxygen under simulated sunlight illumination.

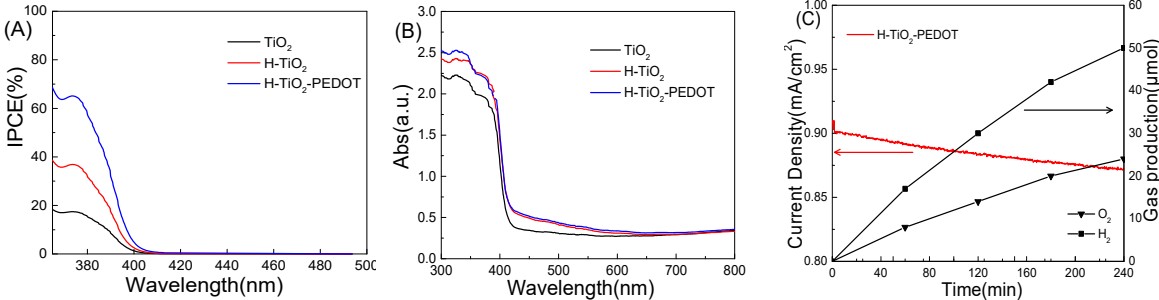

**Figure 6.** The IPCE curves of series photoelectrodes under 0.5 V (vs. Ag/AgCl) (**A**), UV-Vis DRS of series photoelectrodes (**B**), the stability test of H-TiO$_2$-PEDOT under 0.5 V (vs. Ag/AgCl) and corresponding oxygen and hydrogen evolution performance (**C**).

In Table 2, the related research on TiO$_2$ electrodes in recent years is listed. When comparing them, we can see that H-TiO$_2$-PEDOT electrodes presented in this work obtained relatively high PEC performance.

**Table 2.** Statistical list of references.

| Electrode | Light Source | Voltage | Electrolyte | Current Density |
|---|---|---|---|---|
| TiO$_2$ B-NRs [23] | Xe lamp 88 mW cm$^{-2}$ | 1.1 V | 1 M KOH | 0.8 mA/cm$^2$ |
| TiO$_2$ nanorod array [24] | AM 1.5 100 mW cm$^{-2}$ | 0.5 V | 0.5 M NaClO$_4$ | 15 μA/cm$^2$ |
| Carbon Dot/TiO$_2$ Nanorod [25] | Xe lamp 88 mW cm$^{-2}$ | 0 V | 0.1 M NaSO$_4$ + 0.01 M Na$_2$S | 0.35 mA/cm$^2$ |
| H:TiO$_2$ nanotube arrays [26] | AM 1.5G 100 mW cm$^{-2}$ | 0 V | 1 M NaOH | 0.6 mA/cm$^2$ |
| TiO$_2$ nanotubes [27] | UV light 70 mW cm$^{-2}$ | 0.2 V | 1 M KOH | 0.125 mA/cm$^2$ |
| This Work | Simulated sunlight 100 mW cm$^{-2}$ | 0.5 V | 0.1 M NaSO$_4$ + KPi | 0.9 mA/cm$^2$ |

The PEC performance improving the mechanism of H-TiO$_2$-PEDOT nanorod photoanode is shown in Figure 7. Firstly, a nanorod array structure of TiO$_2$ was prepared, which provided a unique route for the photogenerated electron transfer and reduced the recombination rate. In addition, after hydrogen treatment, oxygen vacancies formed on the surface of TiO$_2$ nanorod, increasing the concentrations of free charge carriers. Lastly, a PEDOT layer was deposited on the surface of oxygen vacancy modified TiO$_2$, to inhibit the surface states and improve the separation of photo-induced carriers further by p-n heterojunction formation between PEDOT and TiO$_2$. Thus, more photogenerated holes were transferred to the PEDOT layer and oxidized water, whereas more photogenerated electrons

were transferred to the FTO substrate through the TiO$_2$ nanorod to improve the PEC performance of H-TiO$_2$-PEDOT photoanode.

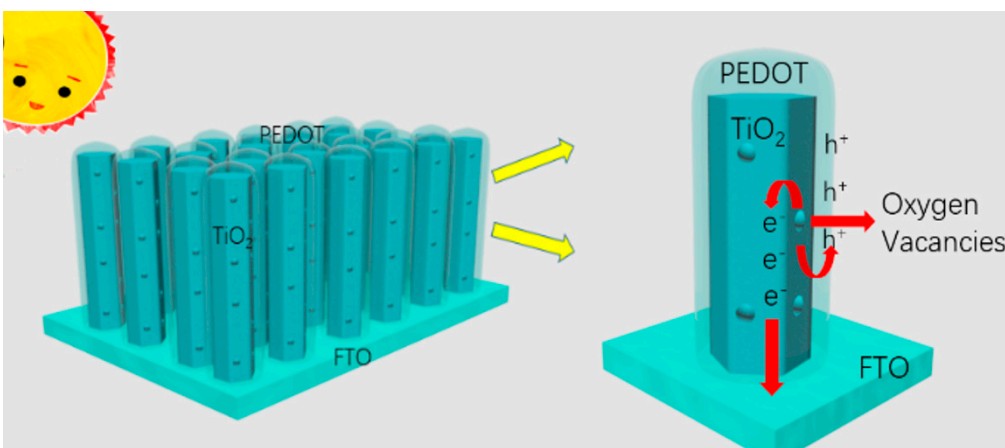

**Figure 7.** Schematic diagram of the mechanism of H-TiO$_2$-PEDOT nanorod thin film under simulated sunlight.

## 3. Materials and Methods

All reagents used in this study were purchased from Aladdin Industrial Corporation (Shanghai, China) with analytical grade. Tetrabutyl titanate, 3,4-ethylenedioxythiophene, and sodium dodecyl sulfonate were not further purified.

### 3.1. TiO$_2$ Nanorod Array and Oxygen Vacancy Modified TiO$_2$ Nanorod Preparation

The TiO$_2$ nanorod arrays were prepared through the solvothermal method. In a typical synthesis, 0.5 mL tetrabutyl titanate was dissolved in 15 mL of hydrochloric acid (36.5%) under continuous stirring, and then 15 mL of deionized water was added for another 5 min to obtain a homogenous solution. The mixed solution was then transferred into a 50 mL Teflon stainless steel autoclave, then two cleaned FTO substrates were immersed into the mixture and kept at 160 °C for 8 h in an oven. After that, the FTO substrates were cleaned with deionized water and then dried under ambient conditions, followed by annealing at 450 °C for 2 h with a ramping rate of 10 °C/min in air in a muffle furnace to obtain TiO$_2$ nanorod array. Then, TiO$_2$ nanorod array was reduced by annealing at 350 °C for 0.5 h with a ramping rate of 10 °C/min in hydrogen conditions, which was denoted as H-TiO$_2$.

### 3.2. PEDOT Preparation

The PEDOT was coated by H-TiO$_2$ nanorod array through electrodeposition method. Typically, 1 mL 3,4-ethylenedioxythiophene (EDOT) and 20 mmol of sodium dodecyl sulfonate (SDS) were dissolved into 200 mL of deionized water under continuous stirring to prepare precursor solution, the deposition process was carried out in a three-electrode system in the above solution. The as-prepared H-TiO$_2$ photoelectrodes, platinum and Ag/AgCl electrode were served as the working, counter, and reference electrodes, respectively. The electrodeposition was carried out using multi-current steps containing 0.01 s of 1 mA of anodic pulse, 0.004 s of 1 mA of cathodic pulse and 0.5 s of 0 A rest current. This above process is termed as one cycle, and 20 cycles were repeated, the obtained electrode was denoted as H-TiO$_2$-PEDOT. Three-electrode system was used to test the H-TiO$_2$-PEDOT stability with an applied bias of 0.5 V (vs. Ag/AgCl). At the same time, oxygen and hydrogen evolution performance were detected by gas chromatography (97900II) regularly.

### 3.3. Characterization

The micromorphology of the prepared photoelectrodes was characterized using a field emission scanning electron microscope (FE-SEM, Ultra 55, Zeiss, Oberkochen, Germany) and a field emission transmission electron microscope (FE-HRTEM, JEM-2100F, Beijing, China). TEM sample was scraping the electrode film into powder and filling the power with alcohol or acetone in a small container. Then a small amount of powder sample was put into it, next, it was placed in an ultrasonic oscillator to vibrate for more than 15 min, and then a copper mesh with supporting film was used to gently pull it out from the solution. The elemental compositions of the photoelectrodes were tested through energy dispersive spectroscopy (EDS, X-max, Oxford Instruments, Oxford, England) and scanning transmission electron microscopy (STEM, JEM-2100F, Tokyo, Japan) mapping. X-ray diffraction (XRD, D/MAX-2500/PC, Rigaku Co., Tokyo, Japan) was used to identify the crystalline structures of the prepared series photoelectrodes. The elementary composition and bonding information of the materials were analyzed using X-ray photoelectron spectroscopy (XPS; Axis Ultra, Kratos Analytical Ltd., Kratos Analytical, Manchester, England). Characterization of the optical absorption properties of a series of electrodes was done by UV-Vis diffuse reflectance (TU-1901, Persee Co., Beijing, China).

### 3.4. PEC Performance Testing

PEC performance measurements were performed in a traditional three-electrode experimental system using Zahner Zennium Pro Electrochemical Workstation (Zahner, Kronach, Germany). The prepared series photoelectrodes, Ag/AgCl (saturated KCl), and a piece of platinum acted as the working, reference, and counter electrodes, respectively. The series photoelectrodes were illuminated under a standard solar simulator (AM1.5G) (LSE341-Zahner, Kronach, Germany). All tests were performed in 0.1 M $Na_2SO_4$ electrolyte. The photocurrent test with time (I-t) curves was measured at a bias potential of 0 V (vs. Ag/AgCl). The linear sweep voltammetry (I-V) curves were measured from −0.5 to 1.5 V (vs. Ag/AgCl) at a scan rate of 0.02 V $s^{-1}$. The IPCE of the photoelectrodes were tested at 0.5 V (vs. Ag/AgCl) bias potential using an IPCE tester (TLS03-Zahner, Germany). Electrochemical impedance spectroscopy (EIS) tests were performed at OCP vs. Ag/AgCl (saturated KCl) over the frequency range between $10^5$ and $10^{-1}$ Hz. Control intensity modulated photocurrent/photovoltage spectroscopy (CIMPS/CIMVS) measured series photoelectrodes with an LED white light source (LSW) from 100 K to 0.1 Hz. The electron transit time ($\tau_r$) and electron lifetime ($\tau_{rec}$) can be obtained by the following Equations:

$$\tau_r = 1/(2\pi\, f_{CIMPS}) \tag{1}$$

$$\tau_{rec} = 1/(2\pi\, f_{CIMVS}) \tag{2}$$

$$\eta(\%) = (1 - \tau_r/\tau_{rec}) \times 100\% \tag{3}$$

where $f_{CIMPS}/f_{CIMVS}$ is the frequency of the minimum imaginary component.

## 4. Conclusions

In this study, PEDOT modified oxygen vacancy-$TiO_2$ nanorod was prepared, oxygen vacancy can improve the charge transfer capacity of $TiO_2$. Meanwhile, the PEDOT could not only serve as the protective layer to inhibit the surface states, but also to fabricate a p-n junction to increase the separation efficiency of the photo-generated electrons and holes. Thus, a near 0.9 mA/cm$^2$ photocurrent of $TiO_2$ nanorod array was achieved after oxygen vacancy and PEDOT co-modification under standard sunlight illumination. Furthermore, the PEC stability test showed that the photocurrent generating by H-$TiO_2$-PEDOT photoanode decays less than 10% of the initial value after 4 h of continuous illumination. Meanwhile, the H-$TiO_2$-PEDOT photoanode can completely split pure water into hydrogen and oxygen under simulated sunlight illumination. Thus, oxygen vacancy and PEDOT co-modification is a promising method for $TiO_2$ photoanode PEC performance improving.

**Author Contributions:** Conceptualization, B.Y.; methodology, H.T., L.W.; experiment and analysis, B.Y., G.C.

**Funding:** This research received no external funding.

**Acknowledgments:** This work was financially supported by the National Natural Science Foundation of China (Grant Nos. 51679227).

**Conflicts of Interest:** The authors declare no conflict of interest.

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
