# Peer review of "Improvement of the Photoelectrochemical Performance of TiO2 Nanorod Array by PEDOT and Oxygen Vacancy Co-Modification"

_catalysts, doi:10.3390/catal9050407_

Round 1

Reviewer 1 Report

In this manuscript, the authors prepared PEDOT modified TiO2/FTO electrode possessing oxygen vacancy and evaluated the photoelectrochemical performance. The content and some results are interesting, however, I personally think that this manuscript should be required minor revisions according to below comments.

1, In abstract part of this manuscript, the authors mentioned that the theoretical limited value of photocurrent of anatase is 1.1 mA/cm2. How was this value calculated? If there is reference paper, the authors should cite in this manuscript.

2, IPCE and APCE measurements are one of the important evaluation methods for photoelectrochemical systems. If possible, the authors should evaluate TiO2 electrodes in this way.

3, I feel that I need to evaluate the stability of PEDOT. In addition, it is also important to confirm whether oxygen generation corresponding to the photocurrent value is observed.

Author Response

Reviewer 1

In this manuscript, the authors prepared PEDOT modified TiO2/FTO electrode possessing oxygen vacancy and evaluated the photoelectrochemical performance. The content and some results are interesting, however, I personally think that this manuscript should be required minor revisions according to below comments.

1, In abstract part of this manuscript, the authors mentioned that the theoretical limited value of photocurrent of anatase is 1.1 mA/cm2. How was this value calculated? If there is reference paper, the authors should cite in this manuscript.

Answer: Thanks for your suggestion, the relative reference was provided in the revised manuscript, please check it. The theoretical limited photocurrent density for pure anatase and pure rutile TiO2 are 1.1 and 1.8 mA/cm2 under solar light illumination, respectively. So, for the mixed phase TiO2 the theoretical limited value should be in between them.

“[2] Liu, C., Dasgupta, N., Yang, P.; Semiconductor nanowires for artificial photosynthesis. Chemistry of Materials, 2013, 26, 415-422.”

2, IPCE and APCE measurements are one of the important evaluation methods for photoelectrochemical systems. If possible, the authors should evaluate TiO2 electrodes in this way.

Answer: Thanks very much for your suggestion. IPCE of the series electrodes were tested under an applied bias of 0.5 V (vs Ag/AgCl) and the results showed in Figure 6(A). It can be seen that the photoelectric conversion efficiency of hydrogen-reduced TiO2 is significantly higher than that of non-reduced TiO2, and after PEDOT loading, the photoanode of H-TiO2-PEDOT showed the highest photoelectric conversion efficiency. This result conforms to the PEC result showed in Figure 5. Because of the limiting of the installation, we regret that we can’t provide the APCE in this time, but we will test it on the further study of this photoanode after we upgrading the installation.

3, I feel that I need to evaluate the stability of PEDOT. In addition, it is also important to confirm whether oxygen generation corresponding to the photocurrent value is observed.

Answer: thanks very much for your suggestions. Figure 6(C) is a photocurrent-time curve measured continuously for 4 hours under 0.5 V (vs Ag/AgCl) external bias voltage. After 4 hours continuous illumination, the photocurrent generating by H-TiO2-PEDOT photoanode decays less than 10% of the initial value, showing an acceptable stability. Meanwhile, the oxygen and hydrogen evolution performance were tested during the PEC stability testing for 4 hours, and the result shows in Figure 6C.

Reviewer 2 Report

The manuscript (catalysts-480906) presents interesting results, which could be of interest to the readers of the Journal. However the presentation must be improved. In this regard, my major comments include:

1.      The introduction should be improved. A large number of scientific articles describe methods for TiO2 modifying. The information presented in the introduction is too general and does not introduce the reader to the subject of the article.

2.      Avoid bulk referencing, e.g. [2-8], should be replaced by a review or book chapter.

3.      Line 28: change “heterojunction[2-8]” to “heterojunction [2-8]”.

4.      Check the entire text of the manuscript for double spaces.

5.      The captions in Figure 3 are too short, the author must provide more information and details enough to clarify the figures.

6.      How many times have the measurements been repeated?

7.      The presented Discussion should be improved. There is no comparison of the obtained results with the literature.

Author Response

Reviewer 2

The manuscript (catalysts-480906) presents interesting results, which could be of interest to the readers of the Journal. However the presentation must be improved. In this regard, my major comments include:

1.          The introduction should be improved. A large number of scientific articles describe methods for TiO2 modifying. The information presented in the introduction is too general and does not introduce the reader to the subject of the article.

Answer: Thanks very much for your suggestion. We have added some modifications to the background section.

“TiO2 has been widely investigated in the past few decades since Fujishima and Honda first reported its potential in the fields of photocatalysis and photoelectrochemistry (PEC) in 1972 [1]. The theoretical limited photocurrent density of anatase and rutile TiO2 are 1.1 mA/cm2 and 1.8 mA/cm2 under solar light illumination, respectively. [2] Limited by its low solar light utilization rate and high photo-generated carrier recombination rate, many modification methods have been researched such as metal doped [3], non-metal doped [4], construct heterojunction [5]. Several elements have been introduced into TiO2 such as Fe [6], S [7] and N [8]. The metal and non-metal doped could narrow the bandgap, extend the light absorption range and increase the charge carrier density to improve its photocatalysis performance. However, the introduction of heterogeneous atoms is likely to cause asymmetric doping or impurities, which would serve as recombination centers for the photo-generated electrons and holes, therefore reducing the photoelectrochemical performance. Many previous research works showed that the formation of surface oxygen vacancy [9-13] could increase the charge carrier density of the semiconductor to improve its PEC performance. Such as Wang et.al [14] obtained a yellowish ZnO with a narrowing band gap by introducing the oxygen vacancies into ZnO crystal, which increased the free charge density of the ZnO, so that feasible the transfer process of the photogenerated charges.

Polymer organic semiconductors, with good film-forming properties, high conductivity, high visible light transmittance and excellent stability, they are widely used in the field of photoelectrode modification. Park et.al [15] use a blend of 100 nm TiO2 scattering particles in PEDOT:PSS solution to fabricate transparent electrode films, when utilized in an organic photovoltaic device, a power conversion efficiency of 7.92% is achieved. Sakai et.al [16] assembled PEDOT and TiO2 Layer-by-Layer for switching of electric conductivity in response to ultraviolet and visible Light. So, PEDOT is a promising material to modify the TiO2 photoanode for its PEC performance improving. [17-20]

Therefore, in this work, we prepared oxygen vacancy modified TiO2 nanorod array photoanode with high charge mobility capacity. Then, a p-type PEDOT layer was covered on the surface of oxygen vacancy modified TiO2 photoanode to inhibit the undesirable surface state and construct a p-n heterojunction to accelerate the separation capacity of photo-generated carriers.[21]”

2.          Avoid bulk referencing, e.g. [2-8], should be replaced by a review or book chapter.

Answer: Thanks very much for your kind reminding. We have revised the bulk referencing.

“Limited by its low solar light utilization rate and high photo-generated carrier recombination rate, many modification methods have been researched such as metal doped [3], non-metal doped [4], construct heterojunction [5].”

3.          Line 28: change “heterojunction[2-8]” to “heterojunction [2-8]”.

Answer: Thanks very much for your reminding. We have changed “heterojunction[2-8]” to “heterojunction [2-8]”. And others similar errors in the manuscript have been revised also.

4.          Check the entire text of the manuscript for double spaces.

Answer: Thanks very much for your reminding. The text space of the manuscript has been changed to double spaces.

5.          The captions in Figure 3 are too short, the author must provide more information and details enough to clarify the figures.

Answer: Thanks very much for your kind suggestion. The captions in Figure have been revised. 

Figure 3. EDS mapping of H-TiO2-PEDOT.(A) Scanning area(B) O Element; (C) C Element; (D) Ti Element; (E) Sn Element; (F) S Element

6.          How many times have the measurements been repeated?

Answer: Thanks very much for your kind reminding, the photocurrent measurements have been repeated at three times and error bar have been introduced into the photocurrent diagrams.

7.      The presented Discussion should be improved. There is no comparison of the obtained results with the literature.

Answer: Thanks for your question. Here is comparison of the obtained results with the literature. In Table 2, the related research on TiO2 based electrodes in recent years is listed. Comparing with each other, we can see that the prepared H-TiO2-PEDOT electrodes have certain advantages and obtain relatively high performance.

Table2 Statistical List of References

Electrode

Light source

Voltage

Electrolyte

Current density

TiO2 B-NRs [24]

Xe lamp 88 mW cm-2

1.1V

1 M KOH

0.8 mA/cm2

TiO2 nanorod array [25]

AM 1.5 100 mW cm-2

0.5 V

0.5 M NaClO4

15 μA/cm2

Carbon Dot/TiO2 Nanorod [26]

Xe lamp 88 mW cm-2

0V

0.1 M NaSO4 + 0.01 M Na2S

0.35 mA/cm2

H:TiO2 nanotube arrays [27]

AM 1.5G 100 mW cm-2

0V

1 M NaOH

0.6 mA/cm2

TiO2 nanotubes [28]

UV light 70 mW cm-2

0.2V

1 M KOH

0.125 mA/cm2

This Work

Simulated sunlight 100 mW cm-2

0.5V

0.1 M NaSO4+KPi

0.9 mA/cm2

Author Response

Reviewer 3

 In this manuscript, authors fabricated TiO2 nanorod array photoelectrode with oxygen vacancies via high temperature reduction in hydrogen gas, which increased the concentrations of charge carriers and surface defect states. They deposited a p-type conductive PEDOT layer was on the surface of the modified TiO2, to inhibit the surface states, which subsequently resulted in p-n heterojunction to further improve the separation of photo-induced carriers. Their research showed that the photocurrent of TiO2 nanorod array increased to near 0.9 mA/cm2 after the co-modification, close the theoretical limited value (1.1 mA/cm2) of anatase TiO2 under sunlight illumination. The reported work could be of good interest to the readers. However, the current manuscript doesn’t meet the publication quality as there are many shortcomings, which need to be addressed (Some of the comments are listed below). Authors need to provide additional experimental data and discussion in the revised manuscript for further consideration. Also, the manuscript needs careful editing as there are numerous confusing sentences, clauses, etc.. The reviewer think the manuscript requires significant improvement before being considered for further assessment. A major revision is recommended. Comments are shown as follows

 Comment 1: Introduction is too short and is not well-documented. It does not cover enough on all aspects of the proposed research work. There are significant work in this field. Authors need to include more important literature on past work highlighting their accomplishment and lack of existing research in the current field to justify their research objective and motivation clearly. Objectives should be written systemically clearly pointing out the hypothesis used in this research. Overall, authors need to pay attention to the introduction as it lacks the quality to meet the journal standard. Following articles are good examples and authors are highly recommended to study and cite the following articles – Chem. Mater. 2006, 18, 16, 3596-3598; Layer-by-Layer Assembled TiO2 Nanoparticle/PEDOT-PSS Composite Films for Switching of Electric Conductivity in Response to Ultraviolet and Visible Light ChemCatChem 2018, 10, 3305, Fabrication of Hierarchical V2O5 Nanorods on TiO2 Nanofibers and Their Enhanced Photocatalytic Activity under Visible Light. ACS Appl. Mater. Interfaces, 2012, 4 (8), pp 4024–4030 “Oxygen Vacancy Induced Band-Gap Narrowing and Enhanced Visible Light Photocatalytic Activity of ZnO J. Am. Chem. Soc. 2015, 137, 8, 2975-2983; Titanium-Defected Undoped Anatase TiO2 with p-Type Conductivity, Room-Temperature Ferromagnetism, and Remarkable Photocatalytic Performance Appl. Phys. Lett. 108, 253302 (2016); PEDOT:PSS with embedded TiO2 nanoparticles as light trapping electrode for organic photovoltaics. Authors can elaborate the discussion on the basis of the above papers showcasing previous study and their importance and limitation.

Answer: Thanks for your question. The introduction has changed as follows and the above literature has been cited.

“TiO2 has been widely investigated in the past few decades since Fujishima and Honda first reported its potential in the fields of photocatalysis and photoelectrochemistry (PEC) in 1972 [1]. The theoretical limited photocurrent density of anatase and rutile TiO2 are 1.1 mA/cm2 and 1.8 mA/cm2 under solar light illumination, respectively. [2] Limited by its low solar light utilization rate and high photo-generated carrier recombination rate, many modification methods have been researched such as metal doped [3], non-metal doped [4], construct heterojunction [5]. Several elements have been introduced into TiO2 such as Fe [6], S [7] and N [8]. The metal and non-metal doped could narrow the bandgap, extend the light absorption range and increase the charge carrier density to improve its photocatalysis performance. However, the introduction of heterogeneous atoms is likely to cause asymmetric doping or impurities, which would serve as recombination centers for the photo-generated electrons and holes, therefore reducing the photoelectrochemical performance. Many previous research works showed that the formation of surface oxygen vacancy [9-13] could increase the charge carrier density of the semiconductor to improve its PEC performance. Such as Wang et.al [14] obtained a yellowish ZnO with a narrowing band gap by introducing the oxygen vacancies into ZnO crystal, which increased the free charge density of the ZnO, so that feasible the transfer process of the photogenerated charges.

Polymer organic semiconductors, with good film-forming properties, high conductivity, high visible light transmittance and excellent stability, they are widely used in the field of photoelectrode modification. Park et.al [15] use a blend of 100 nm TiO2 scattering particles in PEDOT:PSS solution to fabricate transparent electrode films, when utilized in an organic photovoltaic device, a power conversion efficiency of 7.92% is achieved. Sakai et.al [16] assembled PEDOT and TiO2 Layer-by-Layer for switching of electric conductivity in response to ultraviolet and visible Light. So, PEDOT is a promising material to modify the TiO2 photoanode for its PEC performance improving. [17-20]

Therefore, in this work, we prepared oxygen vacancy modified TiO2 nanorod array photoanode with high charge mobility capacity. Then, a p-type PEDOT layer was covered on the surface of oxygen vacancy modified TiO2 photoanode to inhibit the undesirable surface state and construct a p-n heterojunction to accelerate the separation capacity of photo-generated carriers.[21]”

 Comment 2: Fig. 1: XRD peaks in 2theta range ~20-30 should be in plotted. Authors should provide full range XRD data to showcase all the important peaks of TiO2. Typical XRD pattern of TiO2 with the anatase and rutile crystalline phase should appear at 2theta ~25.38 (101); and 2 theta=27.58 (11 0), respectively. Please check these article for your reference - ChemCatChem 2016, 8, 2525. “Mesoporous Titanium Dioxide Nanofibers with a Significantly Enhanced Photocatalytic Activity and Catalysts 2017, 7, 60; doi:10.3390/catal702006” Photocatalytic TiO2 Nanorod Spheres and Arrays Compatible with Flexible Applications”. Cite these articles in the revised version.

- Why there is no reported peaks 2 theta ~25.4 and 27.4 for anatase and rutile TiO2 respectively?

- Include JCPDS# file spectrum of each crystalline form in the graph Fig. 1 Please include standard JCPDS# and spectra in the plot.

- Further discussion of XRD of all 3 materials should be elaborated. crystallinity, crystallite size etc.. should be discussed. Please compare XRD spectra after thermal treatment and H2-reduction. Authors are suggested if the they can check the Crystallite size data from XRD with the results based on TEM images.

Answer: Thanks for your important question. The XRD baselines near 20-30 degree have been added in the revised manuscript, please see Figure 1. It was found that the prepared TiO2 nanorod array showed anatase and rutile mixed phase. The theoretical limited photocurrent density for pure anatase and pure rutile TiO2 are 1.1 and 1.8 mA/cm2 under solar light illumination, respectively. So, for the mixed phase TiO2 the theoretical limited value should be in between them. Base on this result, the theoretical limited photocurrent density for pure anatase 1.1 mA/cm2 in the Abstract has been canceled.

 “The XRD patterns of series samples were shown in Figure 1, all diffraction peaks of the prepared three photoelectrodes can be indexed as rutile-type and anatase-type TiO2 (JCPDS No. 21-1276, JCPDS No. 21-1272) [22,23]. The characteristic diffraction peaks at 2θ = 27.44°, 36.13° and 62.76° corresponded to the (110), (101) and (002) crystal planes of rutile-type TiO2, and the XRD peaks at 2θ=53.97° and 68.79° corresponded to the (105) and (116) crystal planes of anatase-type TiO2. The other characteristic diffraction peaks at 2θ =3 4.96°, 51.76° and 65.94° corresponded to the (200), (211) and (301) crystal planes of SnO2 (JCPDS No. 21-1250), which caused by the FTO conductive glass. So, the prepared TiO2 nanorod array included both anatase and rutile phases.”

 “The peaks of (101) and (110) crystal planes can be clearly observed in the HRTEM insetting in Figure 2E, which is in agreement with the XRD results. Meanwhile, PEDOT layer can be observed at the edge area of TiO2 nanorod.”

Comment 3: “both films appear a distinct nanorod structure with a diameter of 50-100 nm. The cross-sec.” - Please provide high magnification image in Fig. 2

- Average. diameter or diameter distribution measurement of the synthesized TiO2 nanorod should be provided.

Answer: Thanks for your question. We have provided high magnification image in Fig. 2 and provided diameter have distribution measurement of the synthesized TiO2 nanorod.

“The SEM technique was employed to observe the surface morphologies of the series samples, and the results were shown in Figure 2. As presented in Figure 2A-2C, both films appear a distinct nanorod structure with a diameter of 100 nm. The cross-section image of H-TiO2-PEDOT shown in Figure 2D reveal that the TiO2 nanorod growing vertically on the FTO substrate with a height of ~ 1 μm.”

Comment 4: Authors need analyze the surface area of the fabricated materials and provide a detailed discussion on surface area of the materials. Role of surface area and pore size is important considering the photoelectrochemical application of these TiO2 nanorods array.

Answer: Thanks for your question. It is not convenient to measure the specific surface area of thin film photoanode. Moreover, the influence of the preparation process on the surface area is not obvious.

Comment 5: “…Odef), compared to TiO2, the peak area of Odef in H-TiO2 was enhanced after the hydrogen treatment, indicating the increase of oxygen vacancies, which might improve the photo electrochemical performance.”- Can authors calculate % increment of oxygen vacancies? why the XPS spectrum of 2S is noisy?

Answer: Thanks for your question. By integrating the peak area, we obtain the concentration of oxygen vacancies as shown in the figure. Because of the low loading amount of PEDOT, noises can be found on the XPS S2p curve.

Comment 6: Following points need to be taken care of (points are highlighted with yellow)–

- “Surface oxygen vacancy and p-type conductive polymer co-modified TiO2 nanorod array photoanode for its photoelectrochemical performance improving” Please make the title of the manuscript better.

Answer: Thanks for your kind suggestion. The title of this manuscript changed to “Improvement of the photoelectrochemistry performance of TiO2 nanorod array by PEDOT and oxygen vacancy co-modification”.

- please use a different color code to highlight C and S in elemental mapping in EDX if possible.

Answer: Thanks for your kind suggestion. The color code of C and S in elemental mappings have been highlighted.

- “..titanate was dissolved in 15 mL hydrochloric acid36.5%under- concentration?”

Answer: I’m sorry to make an error here. The sentence changes to “0.5 mL tetrabutyl titanate was dissolved in 15 mL hydrochloric acid36.5%under continuous stirring”

- The series photoelectrodes was illuminated under a standard solar simulator”- Band gap energy calculation of the nanorod arrays need to be reported. Please provide the experimental details and results. - Please write a separate material section - list all chemical used and supplier

Answer: Thanks for your questions. Characterization of the optical absorption properties of a series of electrodes by UV-Vis diffuse reflectance (TU-1901, Persee Co., Beijing, China).

 “Figure 6(B) presents the ultraviolet-visible diffuse reflectance result of the series photoanodes. It can be seen that the absorption band edge of pure TiO2 is about 400 nm, because of anatase (band gap 3.2 eV) and rutile (band gap 3.0 eV) mix phase. After hydrogen treatment, an indicated absorption can be found from 400 nm to 600 nm, because of the oxygen vacancy energy level formed on the top of the TiO2 valance band. The light absorption capacity of oxygen modified TiO2 nanorod array dose not changed after the PEDOT out layer loading. Comparing with Figure 6A, there aren’t photocurrent response of H-TiO2-PEDOT photoanode in the wavelength area from 400 nm to 600 nm, indicating that there is no IPCE contribution from oxygen vacancy surface energy level.”

“All reagents used in this study were purchased from Aladdin Industrial Corporation with analytical grade. Tetrabutyl titanate, 3, 4-ethylenedioxythiophene and sodium dodecyl sulfonate were not further purification.”

Comment 7: Authors are recommended to highlight their photo-current performance data in the light of available literature. Please compare and comment on this. How is the morphological robustness of the photo- catalyst?

Answer: Thanks for your question. The comparison of the obtained result with the previous literatures showed in Table 2. Comparing with each other, it shows that the prepared H-TiO2-PEDOT photoanode has excellent PEC performance.

Table 2. Statistical List of References

Electrode

Light source

Voltage

Electrolyte

Current density

TiO2 B-NRs [24]

Xe lamp

88 mW cm-2

1.1V

1 M KOH

0.8mA/cm2

TiO2 nanorod array [25]

AM 1.5 100 mW cm-2

0.5 V

0.5 M NaClO4

15 μA cm2

Carbon Dot/TiO2 Nanorod [26]

Xe lamp

88 mW cm-2

0V

0.1 M NaSO4 and 0.01 M Na2S

0.35 mA/cm2

H:TiO2 nanotube arrays [27]

AM 1.5G 100 mW cm-2

0V

1 M NaOH

0.6mA/cm2

TiO2 nanotubes [28]

UV light 70 mWcm-2

0.2V

1 M KOH

0.125 mA/cm2

The PEC stability of H-TiO2-PEDOT photoanode was tested. Figure 6(C) is a photocurrent-time curve measured continuously for 4 hours under 0.5 V (vs Ag/AgCl) external bias voltage. After 4 hours continuous illumination, the photocurrent generating by H-TiO2-PEDOT photoanode decays less than 10% of the initial value, showing an acceptable stability. Meanwhile, the oxygen and hydrogen evolution performance were tested during the PEC stability testing for 4 hours, and the result shows in Figure 6C.

Comment 8: Please clearly explain the morphological characteristics of the photocatalysts and its correlation with the catalytic properties. Conclusions are not properly written. Some editing is necessary.

Answer: Thanks very much for your kind reminding. In this work, all samples showed a nanorod array structure, so in the manuscript we took more explains on the oxygen vacancy and PEDOT modification. The morphological characteristics of the photocatalysts and its correlation with the catalytic properties explain as follows,

The PEC performance improving mechanism of H-TiO2-PEDOT nanorod photoanode was shown in Figure 7. Firstly, a nanorod array structure of TiO2 was prepared, which providing unique route for the photogenerated electron transfer and reducing the recombination rate. In addition, after hydrogen treatment, oxygen vacancies formed on the surface of TiO2 nanorod, increasing the concentrations of free charge carriers. Lastly, PEDOT layer was deposited on the surface of oxygen vacancy modified TiO2, to inhibit the surface states and improve the separation of photo-induced carriers further by p-n heterojunction formation between PEDOT and TiO2. Thus, more photogenerated holes transfer to PEDOT layer and oxidize water, whereas more photogenerated electrons transfer to the FTO substrate through the TiO2 nanorod to improve the PEC performance of H-TiO2-PEDOT photoanode.

Conclusions are revised as follows:

In this study, PEDOT modified oxygen vacancy-TiO2 nanorod was prepared, oxygen vacancy can improve the charge transfer capacity of TiO2. Meanwhile, the PEDOT could not only serve as the protective layer to inhibit the surface states, but also fabrication a p-n junction to increase the separation efficiency of the photo-generated electrons and holes. Thus, a near 0.9 mA/cm2 photocurrent of TiO2 nanorod array was achieved after oxygen vacancy and PEDOT co-modification under standard sunlight illumination. Furthermore, the PEC stability test showed that the photocurrent generating by H-TiO2-PEDOT photoanode decays less than 10% of the initial value after 4 hours continuous illumination. Meanwhile, the H-TiO2-PEDOT photoanode can completely split pure water into hydrogen and oxygen under simulate sun light illumination. Thus, oxygen vacancy and PEDOT co-modification is a promising method for TiO2 photoanode PEC performance improving.

Round 2

Reviewer 2 Report

The authors improved the manuscript with the suggested information, but some parts of the article still need to be improved:

1.       Explain the abbreviation PEDOT: PSS.

2.       Figure 2.: Figure 2 (F) is missing.

3.       The Materials and Methods chapter should be placed before “Conclusions”.

4.       What equipment was used for the TEM measurement?

Author Response

Reviewer 2:

The authors improved the manuscript with the suggested information, but some parts of the article still need to be improved:

1.           Explain the abbreviation PEDOT: PSS.

Answer: Thanks for your reminding. PEDOT: PSS is the abbreviation of poly 3, 4‐ethylenedioxythiophene: poly styrenesulfonate. We have been added it in the revised manuscript.

2.           Figure 2.: Figure 2 (F) is missing.

Answer: Thanks for your reminding. We had made a mistake. The title of Figure 2 (F) has been deleted in the revised version.

3.       The Materials and Methods chapter should be placed before “Conclusions”.

Answer: Thanks for your comment. The “Materials and methods” has been placed before the “Conclusions” in the revised version.

4.       What equipment was used for the TEM measurement?

Answer: Thanks for your question. The micromorphology of the prepared photoelectrodes were characterized using a field emission scanning electron microscope (FE-SEM, Ultra 55, Zeiss, Germany) and a field emission transmission electron microscope (FE-HRTEM, JEM-2100F).

Author Response

Reviewer 3  

Authors have provided their responses to the reviewer queries. Some of them are valid and they are included in the revised version. The quality of manuscript improved after the 1st revision. However, the reviewer thinks that some more changes are necessary before final decision (see comments below). Hence, another round of revision is recommended. The reviewer would like to see the following changes made in the revised version before being considered for publication. Comments are shown as follows-

Title: “Improvement of the photoelectrochemistry performance of TiO2 nanorod…” 'Photoelectrochemcial ' would be the most appropriate word not ‘chemistry’?

“….its potential in the fields of photocatalysis and photoelectrochemistry (PEC)….” Does the authors refer photoelectrochemical (PEC)?

“….photo-generated electrons and holes, therefore reducing the photoelectrochemical performance…”- Authors should be careful in using the appropriate and consistent word such 'Photoelectrochemical' in the manuscript. It would make the reader a bit confusing to see word like ‘photoelectrochemistry’ , ‘photoelectrochemical’..

Answer: Thanks for your kindly suggestion. We have corrected them in the full text. In the revised manuscript PEC means photoelectrochemical.

define full form- “..particles in PEDOT:PSS solution..” at the first time.

Answer: Answer: Thanks for your reminding. PEDOT: PSS is the abbreviation of poly 3, 4‐ethylenedioxythiophene: poly styrenesulfonate. We have been added it in the revised manuscript.

 “….heterojunction to accelerate the separation capacity of photo-generated carriers.[21]…” – The reviewer think that ref 21 would be most appropriate here Line 36: –“….. many modification methods have been researched such as metal doped [3], non-metal doped [4], construct heterojunction [place ref#21 here]..".

Answer: Thanks very much for your reminding. We made a change that ref [21] is appropriate for Line 36, and it has been changed to ref [5].

Fig. 1: XRD plot should be more clear and prominent. some important peaks are hardly visible. Could the authors explain why the anatase and rutile peaks 2theta ~25.4 and 27.4 are not intense in the spectra? Please index the peak with crystal planes corresponding to anatase and rutile TiO2 in the plot. And same applies for SnO2 in the XRD spectra.

Answer: Thanks for your question. The XRD patterns of series samples were shown in Figure 1, all diffraction peaks of the prepared three photoelectrodes can be indexed as rutile-type and anatase-type TiO2 (JCPDS No. 21-1276, JCPDS No. 21-1272) [21,22]. The characteristic diffraction peaks at 2θ =36.08°, 54.32, 62.74 and 69.78° corresponded to the (101), (211), (002) and (112) crystal planes of rutile-type TiO2, and the XRD peaks at 2θ=63.68° corresponded to the (204) crystal planes of anatase-type TiO2. The other characteristic diffraction peaks at 2θ =26.57°, 37.76°, 51.75° and 65.74° corresponded to the (110), (200), (211) and (301) crystal planes of SnO2 (JCPDS No. 46-1088), which caused by the fluorine doped tin oxide (FTO) conductive glass. So, the prepared TiO2 nanorod array included rutile phases and a little anatase phases. The TiO2 nanorod array preparing method in this work was referred to Liu’s work,[23]. The vanished peaks for anatase and rutile TiO2 at 25.4° and 27.4° on the XRD curves maybe attribute to the crystal face inhibition effect of the oriented growth nanorod structure, which results is similar to Liu’s work. [23]

Fig 2: where is fig 2F? : Fig 2 (E) STEM mapping of H-TiO2-PEDOT. (F) STEM mapping of H-TiO2PEDOT. Are they same? Please correct this.

Answer: Thanks for your kind reminding. We had made a mistake here and the title of Figure 2 (F) has been deleted in the revised version.

Line: 133-“…The peaks of (101) and (110) crystal planes can be clearly observed in the HRTEM 133 image inset in Figure 2E, which is in agreement with the XRD results… "- check the fig. # ?

Answer: Thanks for your comment. We had made a mistake here and we revised the sentence as follows after check the figure number. The Figure 2E has been changed to Figure 2D in the revised manuscript.

Line 130: "As presented in Figure 2A-2C, both films appear a distinct nanorod structure with a diameter of 100 nm "-The meaning of the above sentence is not clear. Also, provide the diameter distribution or ave. diameter of the nanorods. It is clear that all the nanorods don’t have exactly 100 nm diameter.

Answer: Thanks for your suggestion. The SEM technique was employed to observe the surface morphologies of the series samples, and the results were shown in Figure 2. As presented in Figure 2A-2C, TiO2, H-TiO2 and H-TiO2-PEDOT appear a distinct nanorod structure. The cross-section image of H-TiO2-PEDOT shown in Figure 2D reveal that the TiO2 nanorod growing vertically on the FTO substrate. The nanorods are tetragonal in shape with square top facets, the expected growth habit for the tetragonal crystal structure. The nanorods are nearly perpendicular to the FTO substrate. After 8 h of growth, the average diameter and length, as determined from SEM, were 90±20 nm and 1±0.2 µm, respectively.

Line 146: “..simultaneously, which phenomenal is corresponding to the STEM mapping showed in 146 Figure 2E…” - check fig # 2E?

Answer: Thanks for your reminding. We had made a mistake here and the title of Figure 2 (F) has been converted to Figure 2(E) in the revised manuscript.

Fig. 4A- Please define Vo. Also, in case of H-TiO2 PEDOT and H-TiO2 samples, the Odef area appears to be significantly different but the values of Vo are almost same, 43.74% and 43.77% respectively? Please double check the graphs.

Answer: Thanks for your question. Oxygen vacancy concentration refers to the proportion of oxygen vacancy peak area to total oxygen peak area. Because XPS can only read the distribution of surface elements, the peak area of oxygen (O 1s, Olat, Odef ) becomes smaller after PEDOT loading, but the relative content is credible.

Line 211: “,absorption band edge of pure TiO2 is about 400 nm, because of ….” .Mention the band gap energy of pure TiO2?

Answer: Thanks for your question. This sentence has been revised.

“It can be seen that the absorption band edge of pure TiO2 is about 400 nm, because of anatase (band gap 3.2 eV) and rutile (band gap 3.0 eV) mix phase.”

Authors included HR-TEM discussion. Include TEM instrument details and brief sample preparation in characterization section? Also L-127.

Answer: Thanks for your comment. The TEM instrument and the brief sample preparation has been added in the characterization section.

“The peaks of (101) crystal planes can be clearly observed in the HRTEM image inset in Figure 2D, which is in agreement with the XRD results. The micromorphology of the prepared photoelectrodes were characterized using a field emission scanning electron microscope (FE-SEM, Ultra 55, Zeiss, Germany) and a field emission transmission electron microscope (FE-HRTEM, JEM-2100F). We scraped the electrode film into powder and filled the power with alcohol or acetone in a small container. Then a small amount of powder sample put into it, and then put it in an ultrasonic oscillator to vibrate for more than 15 minutes, and then use a copper mesh with supporting film to gently pull it out in the solution.”